# May Female Autism Spectrum Be Masked by Eating Disorders, Borderline Personality Disorder, or Complex PTSD Symptoms? A Case Series

**DOI:** 10.3390/brainsci14010037

**Published:** 2023-12-30

**Authors:** Barbara Carpita, Benedetta Nardi, Cristiana Pronestì, Francesca Parri, Federico Giovannoni, Ivan Mirko Cremone, Stefano Pini, Liliana Dell’Osso

**Affiliations:** Department of Clinical and Experimental Medicine, University of Pisa, 56126 Pisa, Italy; barbara.carpita@unipi.it (B.C.); cristianapronesti@gmail.com (C.P.); francyparri@icloud.com (F.P.); f.giovannoni10@gmail.com (F.G.); stefano.pini@unipi.it (S.P.); liliana.dellosso@unipi.it (L.D.)

**Keywords:** autism, autism spectrum disorder, PTSD complex, borderline personality disorder, eating disorder, autism female phenotype

## Abstract

**Introduction:** The prevalence of Autism Spectrum Disorder (ASD) is four times higher in males than females; however, females are significantly more likely to go undiagnosed due to the existence of a “female autistic phenotype”, a manifestation unique to females that conflicts with conventional, masculine conceptualizations of ASD. Furthermore, subthreshold autistic traits, which exert a significantly negative impact on quality of life and represent a vulnerability factor for the development of other psychopathological conditions, may remain even more under-recognized. Subsequently, many women with ASD may never receive a diagnosis or any resulting care, with serious consequences for their health. **Aims:** We aimed to describe two brief cases in order to confirm the diagnostic difficulties that ASD female undergo during their clinical evaluation and the possible alternative phenotype that they can manifest. **Methods:** We reported the cases of two young women on the autism spectrum that came to clinical attention only after the development of severe symptomatology attributed to other mental disorders, overlooking the presence of underlying autism spectrum features and a brief résumé of the literature on this topic. **Results:** These cases confirm the need for a timely and proper identification of females on the autism spectrum in order to prevent complications and improve the outcome. **Conclusions:** Research on gender differences could lead to a reexamination of the sex ratio in the prevalence of ASD and provide a better understanding of several psychiatric conditions that are frequently diagnosed in women, supporting the neurodevelopmental approach to psychopathology.

## 1. Introduction

Autism spectrum disorder (ASD) is a neurodevelopmental disorder characterized by challenges with social reciprocity, social communication, adaptability, and sensory processing [1]. Individuals with ASD often face the risk of experiencing a series of behavioral, emotional, social, and professional challenges, such as maintaining romantic and friendship relationships, fitting into a work environment, and adapting to the norms imposed by the sociocultural context in which they find themselves [2]. By identifying needs and appropriate interventions, expanding access to services, decreasing self-criticism, and fostering a positive sense of identity, early detection of female ASD can help reduce some of the risks and improve the quality of life [3,4,5,6,7].

Historically, the prevalence of ASD is reported to be approximately four times greater in males than in females [8,9,10]; however, the recent literature has highlighted how females with ASD have a significantly higher chance than males to remain undiagnosed [11]. One proposed explanation for the diagnostic bias against females with ASD finds its ground in recognition of the existence of a “female autism phenotype”, a manifestation of autism that is unique to women and that conflicts with the conventional, male-based conceptualizations of ASD [1,12,13,14]. Females with ASD have consistently been reported to show less repetitive and stereotyped behavior [15] and are less likely to experience externalizing behaviors like hyperactivity/impulsivity and conduct issues, whereas they are more susceptible to internalizing problems like anxiety, depression, and eating disorders [14,16]. Indeed, growing evidence supports the existence of the female phenotype, reporting how, compared to men, ASD females exhibit higher levels of social drive and a larger ability for conventional social connections [17,18,19].

Moreover, it has been proposed that one of the main features of the female phenotype is the ability to “camouflage” social impairments in a variety of social settings [20]. Indeed, in order to appear socially competent, ASD females have been reported to perform many camouflaging techniques, such as hiding some types of behaviors that could be considered socially inappropriate or explicitly performing other behaviors considered neurotypical [21,22,23]. These behaviors can be picked up by mimicking peers or even by mechanically analyzing various media, and when performed consistently from a young age, they can become increasingly complicated over time and be somewhat unintended [24]. In certain instances, it could also result in the creation of a new identity—sometimes called a “mask”—that the person puts on in social situations [22]. The idea that the employment of camouflaging methods contributes to the underdiagnosis of ASD in females appears, thus, reliable, given that social camouflaging is often used by high-functioning subjects and that there is a simultaneous decrease in the gender gap in ASD diagnosis and lower IQs [25,26]. Subsequently, females with ASD who do not have a language or intellectual disability would probably be diagnosed with other conditions, which would negatively affect both the course of their illness and the effectiveness of their therapy [27].

Furthermore, subthreshold autistic traits may remain even more underrecognized in this population. On the other hand, in the recent literature, increasing attention has been focused on the issue of subthreshold autism conditions, which have been firstly identified in first-degree relatives of ASD subjects and known in this population under the name of “broad autism phenotype” (BAP), encompassing all those autistic-like features such as inflexibility, tendency toward isolation, aloof personality, and strong and narrow interests, which, for number or severity, fail to meet the diagnostic criteria for a full-blown ASD diagnosis [28]. While the concept of BAP stresses the common genetic basis between under and full-threshold autism spectrum symptoms, the presence of subthreshold autistic traits seems to be continuously distributed from the general to the clinical population [29]. The importance of detecting subthreshold autistic traits lies in the fact that they seem to exert a significantly negative impact on quality of life, being also a vulnerability factor for the development of other psychopathological and somatic conditions [28,29,30]. Noticeably, studies in this field showed that a similar suicidality risk between subjects with ASD and subthreshold autistic traits in both cases significantly increased compared to controls without autism spectrum conditions [31,32].

In this framework, many authors have recently hypothesized a possible reconceptualization of some feeding and eating disorders (FEDs) and, in particular, anorexia nervosa (AN) as the female phenotype of ASD, where the restricted and intense interests revolve around food [33,34,35]. Another disorder mostly diagnosed among females and that has recently been suggested to be a possible manifestation of ASD in females is borderline personality disorder (BPD) [36]. Indeed, despite the apparent differences between the two disorders, significant overlaps have been identified in more recent publications. In particular, both disorders share the tendency to intense relationships and superficial friendships, high rates of self-injurious behaviors, impairment in verbal and non-verbal communications, social functioning, and emotional regulation, as well as difficulties in the theory of mind [24,37,38,39,40,41,42,43,44,45,46]. Moreover, because of their social challenges and impaired socioemotional reciprocity, subjects with ASD are more likely to experience bullying, rejection, and other socially stressful or even traumatic situations, resulting in the development of trauma and stress-related symptoms [27,31]. In particular, these individuals with heightened susceptibility may develop a peculiar form of post-traumatic stress disorder (PTSD), extremely similar to the presentation of BPD, the complex PTSD (cPTSD), following the experience of traumatic events of lesser severity than those listed in DSM-5 criteria A [47,48].

Ultimately, females are far more likely than males to have an undetected ASD because their problems are sometimes mislabeled or overlooked completely [11]. As a result, many females who, upon a skilled assessment, would fully meet the diagnostic criteria for ASD never receive a diagnosis or the possible assistance that accompanies it, and even when they are found, they are diagnosed later than their male counterparts and receive less support in the meanwhile [49]; indeed, the majority of programs in mental health are focused on the care of first-episode of psychosis, which is a far more common male presentation. Moreover, teachers underreport autistic features in their female students, which, in turn, need more severe autistic symptoms and more cognitive and behavioral deficits than male students do in order to be recognized [25,50,51]. These prejudices are unavoidable, and they appear to have been particularly prevalent among those who are not experts in diagnosing neurodevelopmental disorders but who are, nonetheless, powerful gatekeepers to pertinent treatments for individuals with ASD [52,53]. The lack of research on this gender prejudice has finally been recognized as one of the main key problems in the autism community and has important implications for the health and well-being of girls and women on the spectrum [54,55].

In this work, we reported the cases of two young women within the autistic spectrum, where spectrum means the set of typical symptoms described by the DSM-5, atypical symptoms, clusters of subclinical symptoms, personality traits, and related behavioral manifestations, without intellectual impairment that came to clinical attention only after the development of severe symptomatology, which was attributed to personality disorders, FEDs, or trauma-related disorders, overlooking the presence of underlying autism spectrum features.

## 2. Methods

The subjects were recruited from inpatient afferents at the Psychiatric Clinic of the University of Pisa and did not report any language or intellectual disabilities that made it difficult to complete the examinations.

Early symptoms that may have been present during childhood were assessed with the help of parents and through lifetime self-report questionnaires. The evaluation of early symptoms and the provision of a comprehensive psychobiography appears particularly relevant in the evaluation of females with ASD. In fact, some typical characteristics of ASD, such as deficits in verbal and non-verbal communication, can subsequently be masked and, therefore, difficult to detect due to the acquisition of camouflaging strategies.

During the hospitalization, they underwent daily observation and assessment from a trained psychiatrist and were evaluated with the following self-report questionnaire.

### 2.1. Adult Autism Subthreshold Spectrum (AdAS Spectrum)

The AdAS Spectrum is a self-report questionnaire designed to assess the wide range of autism spectrum manifestations in adults who do not have language or intellectual disabilities. It consists of 160 dichotomous items organized into seven domains, which explore *Childhood and adolescence, Verbal communication, Nonverbal communication, Empathy, Inflexibility and Adherence to Routine, Restricted interests and rumination,* and *Hyper- and Hyporeactivity to Sensory Input*. The questionnaire showed great psychometric properties, excellent internal consistency for the total score (Kuder–Richardson’s coefficient = 0.964), convergent validity with other dimensional measures of ASD, and a diagnostic threshold score of 70 [56,57].

### 2.2. Camouflaging Autistic Traits Questionnaire (CAT–Q)

The CAT-Q is a questionnaire developed by Hull et al. to assess the dimension of camouflaging behaviors, which is also available in an Italian version. Both versions showed great internal consistency with a Cronbach’s alpha of 0.904 in the Italian version and test–retest reliability and convergent validity with other measures of ASD [58,59]. The questionnaire consists of 25 items organized on a seven-point Likert scale and divided into three domains investigating *Compensation, Masking,* and *Assimilation* behaviors.

### 2.3. Eating Disorder Inventory 2 (EDI-2)

The EDI-2 is a self-report questionnaire that measures psychological traits and symptom clusters relevant to the assessment of individuals with eating disorders. It consists of 91 items rated on a six-point scale from never to always organized in 11 domains, such as *Drive towards thinness, Bulimia, Body dissatisfaction, Ineffectiveness, Perfectionism, Interpersonal distrust, Interoceptive Awareness, Maturity fears, Asceticism, Impulse Regulation,* and *Social Insecurity*. The EDI-2 is vastly used in FED research and has been validated for both clinical and general populations [60].

### 2.4. The ORTO-15

The ORTO-15 is a self-report questionnaire used for the assessment of eating behaviors ascribable to Orthorexia Nervosa (ON). It consisted of 15 items rated on a four-point Likert scale and recognized two validated threshold scores of <35, which displayed greater specificity, and <40, which displayed a greater sensitivity [61,62].

The choice to include the ORRTO-15 in the assessment of the first patient was guided by the increasing body of research that confirms important symptomatologic overlap between ON and AN. Specifically, ON similarities to AN are sparking discussion on whether the first is a distinct disorder or a subtype of AN that parallels the growth of the modern ideal of healthy eating, which is progressively displacing the ideal of thinness that was prevalent in the 1980s and 1990s. In addition to the possibility of dramatic weight loss, ON and AN also share high levels of anxiety, a strong drive to exercise control, and a tendency toward perfectionism.

### 2.5. Trauma and Loss Spectrum—Self Report (TALS-SR)

The TALS-SR questionnaire investigates the lifetime experience of a range of traumatic events or losses as well as symptoms, behaviors, and personological traits that might represent manifestations or risk factors for the development of a stress-related disorder. It consists of 116 dichotomous items organized into nine domains, exploring *Loss events, Grief reactions, Potentially traumatic events, Reactions to losses or upsetting events, Re-experiencing, Avoidance and numbing, Maladaptive coping, Arousal,* and *Personal characteristics/risk factors*. The questionnaire showed great psychometric properties overall [63,64].

## 3. Case Presentations

### 3.1. Case X.Y.

X.Y. is a 23-year-old girl who lives in a northern Italian city with her sister. In the medical history of her family, the occurrence of any mental disorder was not reported; however, her father was reported to have experienced a language development delay similar to hers.

Since childhood, she was described as irritable and ruminative, a perfectionist with the need for order and control, hypersensitive to judgment and criticism, and with a great fear of reptiles. She reported to have practiced acts of bullying, being prone to isolation, having distrust toward others, having few friendly relationships, and not being interested in sentimental ones. She also manifested cluster B personality traits such as impulsivity, tendency to substance abuse, self-harm, and novelty seeking.

The onset of the psychopathological picture, reported by the parents, dates back to the age of 12 when the patient carried out restrictive and selective eating behaviors with progressive weight loss of an unspecified amount. These behaviors continued throughout adolescence and were also present at the time of the interview, with a BMI of 16.8.

At the age of 15, X.Y. began to carry out self-harming acts, including self-cutting and burning with cigarettes, aimed at compensating for the sense of emptiness and boredom experienced, and at 17, she began using cannabinoids for recreational purposes.

From the age of 18, X.Y. reported the onset of episodes of increased free anxiety with subcritical episodes characterized by symptoms of depersonalization and derealization and, one year later, the appearance of doubts regarding her sexual orientation.

At the age of 20, after the failure of a university exam, she experienced full-blown panic attacks with neurovegetative and cardiorespiratory symptoms, which she managed to interrupt through the implementation of self-harming gestures. At the same time, food restriction behaviors intensified, associated with strict control of the number of calories in meals, intense physical activity, and the use of diuretics and laxatives.

When she was 21, she began a psychodynamic/psychoanalytic psychotherapeutic treatment with reported partial benefit, during which she was advised to contact a psychiatric service. After the first contact with the psychiatric service, she was given a therapy based on benzodiazepines, which she then reported having abused, reaching the assumption of 10 mg of Lormetazepam per day.

A few months later, in the apparent absence of stressful life events, X.Y. experienced an increase in free-floating anxiety with critical episodes and cardiorespiratory symptoms, interpersonal conflict with friends and family, and an increase in substance abuse (cannabinoids) for self-therapeutic purposes in order to control anxious symptoms and to counteract the sense of inadequacy and hypersensitivity to the judgment of others.

As time passed, the symptomatologic picture worsened with significant weight loss (BMI: 15) and the appearance of ideas of reference and persecution, which ultimately evolved toward a delirium of substitution aimed at her parents and the resurgence of restrictive eating habits. Interestingly, during this time, the girl had a variable and discontinuous insight into her disorder, and sometimes, she recognized her beliefs as erroneous, unfounded, and pathological.

The parents then contacted a new psychiatric specialist without informing her, which resulted in a heated family argument and culminated in a visit to the emergency care from which X.Y. was, however, discharged after a short time, as the criteria for compulsory health treatment under the regime of hospitalization did not subsist. X.Y. then moved away from the family unit and went to a house they owned, interrupting all contact with her parents and sister. The latter, alarmed, contacted the local police, who escorted the girl to the local emergency room and, subsequently, to the service of psychiatric diagnosis and treatment, where she was subjected to voluntary hospitalization and diagnosed with “borderline personality disorder and anorexia nervosa”. During hospitalization, toxicological screening was performed, highlighting a positivity for cannabinoids, and a therapy based on Olanzapine up to 20 mg/day and Delorazepam up to 2 mg/day was set with partial clinical benefit.

Due to the persistence of anxious symptoms, substance abuse behaviors, and eating disorders, the parents, in agreement with the patient, contacted our psychiatric clinic to carry out a new hospitalization. During the clinical interview, ruminative ideas emerged regarding the meaning of life, her value as a person, her sexuality, and her identity. She also showed ideas of persecution and reference, suspecting for some months that there were plots against her and that everyone was watching her.

During the stay, she was assessed with the AdAS Spectrum questionnaire, reporting a score of 77/160, above the threshold (of 70) for a potential clinical diagnosis of ASD (see Table 1). She was also assessed with the CAT-Q, reporting a score of 110/175 above the threshold (of 100) for significant camouflaging behaviors (see Table 2), the ORTO-15 reporting a score of 40/60, and the Eating Disorder Inventory-2 (EDI-2) reporting a score of 139/273 (see Table 3). During the clinical interview, the camouflaging strategies implemented by the patient were explored in depth. In particular, she reported having a series of clothes and accessories specifically designed to appear appropriate to the situation in which they would be worn, for example, clothes for the library, for school, or for the bar. She also reported that she remembered, through the questionnaire, how, during childhood, she had committed herself to studying facial expressions and non-verbal communicative movements, trying to copy those seen by children at school in various situations. Finally, she reported making a conscious effort during conversations to maintain a balance between talking and listening and to fill the gaps with verbal encouragers.

### 3.2. Case W.Z.

W.Z. is a 22-year-old Polish girl of Lebanese origin, studying biology classes at university, working as a waitress, and living alone. The medical history of her family reported the occurrence of mental disorders; specifically, her grandmother suffered from an affective disorder, and two uncles from alcohol use disorder.

She describes herself as reserved, introverted, and sensitive to stress and reassurance, with a strong interest in life sciences. She also manifested difficulties in social relationships, alexithymia, and emotional dysregulation. She also reported, since childhood, intense anxiety in school-related situations, which eventually led to emotional outbursts.

The onset of the psychopathological picture is reported to date back to the age of 12 when, following stressful life events at school and in the family, she developed symptoms characterized by a mood oriented toward the negative pole with social withdrawal, apathy, abulia, reduction in the hedonic volitional drive, concentration deficit, high internal tension, subcritical increases in the anxiety levels, which culminated in self-cutting behaviors and suicidal thoughts.

Starting from the age of 14, W.Z. began an occasional abuse of substances, including marijuana, amphetamines, cocaine, LSD, and opioids, and daily use of alcohol and THC. Due to the progressive worsening of the symptoms, at the age of 16, she made her first contact with psychiatric services, where she was given a therapy based on Citalopram up to 20 mg/day and later changed to Escitalopram up to 20 mg/day, with reportedly little clinical benefit.

At the age of 18, due to the persistence of the symptoms, she turned to a new specialist who stopped the previous therapy and started a new one based on Agomelatine up to 25 mg/day and Lamotrigine up to 50 mg/day, which was interrupted after two weeks due to the appearance of purpura.

Over the years, she turned to various specialists who prescribed numerous therapies based on antidepressants (Sertraline, Venlafaxine, Citalopram, and Escitalopram) and Pregabalin at unknown doses, with reportedly little benefit.

At the age of 20, after experiencing the death of a dear friend of hers from an overdose, she developed symptoms characterized by depressed mood, self-devaluation, distrust, recurring nightmares, increases in anxiety levels, episodes of derealization and depersonalization, emotional numbing, increase in substance abuse behaviors, and self-cutting, which culminated in a suicide attempt by Lamotrigine ingestion, which resulted in her first hospitalization at the local psychiatric diagnosis and treatment service, from which she was discharged with a diagnosis of “borderline personality disorder and substance abuse” after a month with a therapy based on Sertraline up to 75 mg/day, Trazodone up to 75 mg/day, Lorazepam up to 2 mg/day, Zolpidem up to 10 mg/day, and Pregabalin up to 150 mg/day.

In the following months, she managed to maintain sufficient psycho-affective compensation, by virtue of which she decided to autonomously suspend the therapy.

At the age of 21, following stressful vital events, W.Z. developed a clinical picture characterized by critical increases in anxiety levels of cardiorespiratory nature and crying fits with occasional self-cutting behaviors and persistent substance abuse for which she once again turned to a psychiatrist who suggested a therapy with Quetiapine up to 200 mg/day, Venlafaxine up to 75 mg/day, and Pregabalin up to 150 mg/day, with little benefit reported. Substance abuse behaviors also persisted with daily consumption of THC, synthetic opioids, and approximately a bottle of wine a day.

A few months later, the symptoms worsened with a decline in affective tone, poor hedonic volitional drive, feelings of emptiness, suicidal ideation, panic attacks, and daily abuse of mephedrone, for which she moved to her grandmother in Italy and reached our clinic where she was given a therapy based on Valproic Acid up to 1000 mg/day, Paroxetine up to 20 mg/day, Aripiprazole up to 15 mg/day, Delorazepam up to 2 mg/day, and Pregabalin up to 75 mg/day, with partial clinical benefit. At the age of 22, following a loss event, W.Z. experienced a flare-up of the index picture, for which she underwent hospitalization in our ward. During the hospitalization, she was treated with Valproic Acid up to 1000 mg/day, Lithium Solphate up to 124.5 mg/day, Paroxetine up to 20 mg/day, Aripiprazole up to 15 mg/day, Lorazepam up to 2 mg/day, and Perphenazine up to 8 mg/day.

During her stay, she was assessed with the AdAS Spectrum questionnaire, reporting a score of 73/160, above the threshold (of 70) for a potential clinical diagnosis of ASD (see Table 4). She was also assessed with TALS-SR, reporting a score of 60/116 (see Table 5), and the CAT-Q, reporting a score of 82/175 (see Table 6). During the clinical interview, the camouflaging strategies implemented by the patient were explored in depth. In particular, she reported actively trying to reduce her fidgeting and stimming movements and making an effort to intentionally maintain eye contact or the appearance of eye contact, for example, by looking at the interlocutor’s forehead. She also reports that she is used to smiling when someone is talking to her because she has learned over time that it makes her appear less rude and more approachable.

## 4. Discussion

We examined the cases of two young women without intellectual disability whose symptoms, despite the underlying presence of an autism spectrum, were initially misdiagnosed as manifestations of eating or trauma-related disorders.

Both girls reported an AdAS Spectrum score significant for a potential clinical diagnosis of ASD; however, the presence of ASD or subclinical autistic features was never recorded. Recently, a growing body of research highlighted how ASD females have a significantly higher chance than males to remain undiagnosed due to the presence of a female phenotype unique to women, which conflicts with the conventional male-based conceptualizations of ASD [1,11,12,13,14], their stronger social drive [17,18,19], and their ability to “camouflage” social impairments [20]. Indeed, in both cases presented, X.Y. and W.Z., when appropriately investigated, reported significantly high scores in the tests evaluating the adoption of camouflaging behaviors. Camouflaging strategies have been reported to be one of the central elements of the female phenotype, developed due to greater social pressure and obtained due to an intentional and prolonged self-didactic process, focused on careful observation of peers, reading novels and books, and on imitation of television characters, sometimes carried out unconsciously by the patient [20,21,22,23,24]. Not only the subjects’ risk of remaining undiagnosed but additionally, as several authors pointed out, due to the diagnostic criteria for ASD based on the usual male presentations [1,12,13,14], females without language or intellectual disability may receive a different diagnosis with a detrimental effect on how the illness progresses and how well the therapy works [27,54].

In the first case, even though X.Y. reported throughout childhood and adolescence significative autistic traits, she came to clinical attention only after the development of severe affective symptomatology and significant weight loss. Interestingly, the recent literature has highlighted how AN could represent a female presentation of the autism spectrum [33,34,35]. The hypothesis of a possible correlation between AN [65] and ASD is fueled by the evidence of a familiar aggregation for these conditions, as well as by the greater presence of some typically autistic traits in patients with AN and in parallel with abnormal eating behaviors in patients with ASD [66,67]. Some of the first studies looking for this correlation were carried out by Gillberg, who did not fail to notice how AN subjects often showed typical traits of ASD, such as insistence on sameness and compromised social interactions [68]. Over time, the hypothesis of a correlation between AN and ASD has been enriched by the results of numerous studies that have analyzed the similarities between the two conditions. In particular, AN is frequently described as rigidity in set-shifting tasks and global processing, as well as greater attention to detail, traits also typical of the autism spectrum [69]. This is accompanied by high social anhedonia and reduced emotional awareness [70], impaired emotional intelligence [71], and poor emotional processing capacity, including a deficit in the attribution of moods to facial expressions [72]. Even Theory of Mind, the impairment of which is one of the most important features of ASD, seems to be impaired in AN, especially when dealing with advanced tasks [73].

In this framework, following the recognition of the existence of a female phenotype for autism, AN could be considered a specific manifestation of the same, previously unrecognized due to the gender bias present in the diagnostic criteria of ASD, in which the restricted interests and repetitive behaviors typical of autism would be focused on food and diet [33,74,75]. Interestingly, this conceptualization could explain the diagnostic gender gap present in both disorders, namely, the surprisingly higher prevalence of FEDs among female individuals and, conversely, of ASD among male individuals.

In the second case presented, W.Z. developed a symptomatology bordering between the diagnosis of BPD and that of a trauma and stress-related disorder. BPD is a condition that is primarily diagnosed in females, and that shows a significant symptomatologic overlap with many autistic characteristics, such as decreased empathy and social–emotional reciprocity, trouble controlling emotions, altered reactivity and reactions to stimuli or outbursts of anger, and an increase in self-harming and/or suicidal ideation [24,37,38,39,40,41,42,43,44,45,46]. Additionally, an increased prevalence of ASD or subthreshold autistic traits is frequently reported among patients with BPD, while, in turn, ASD patients show a greater frequency of BPD. Particular consideration should be given to the role of trauma and stress-related psychopathology in order to gain a better understanding of the potential connection between these two disorders [76]. In particular, patients with BPD are known to typically have a history of traumatic occurrences, and, in parallel, emerging evidence suggests that ASD, or subthreshold autistic traits, may potentially be risk factors for developing trauma and stress-related disorders [77]. In particular, these individuals with heightened susceptibility can undergo the development of cPTSD, a peculiar form of PTSD extremely similar to the presentation of BPD [48,54]. cPTSD is characterized by the presence, together with the main symptoms of PTSD, of self-organization disorders such as severe and persistent problems in affect regulation, beliefs about the self as belittled, defeated, or worthless, difficulties in sustaining relationships, and feeling close to others. Moreover, previous research has emphasized the parallels between the clinical picture of cPTSD and BPD, which has led to the hypothesis that individuals with cPTSD might frequently be diagnosed with BPD, and, consequently, female ASD may often be misdiagnosed with BPD if they present a history of even mild traumatic experiences [27].

In this context, given the expanding incidence of ASD, the hunt for environmental risk factors is becoming increasingly relevant. One such issue that has drawn attention from researchers is migration with low vitamin D levels and ethnicity proposed as potential explanatory processes [77,78]. However, the quality, sample sizes, and definitions of autism used in prior research on the relationship between migration and the risk of ASD differ significantly and have shown contradictory results, such as higher, comparable, and even lower risk of autism in migrant children [78,79,80,81,82,83,84,85,86]. Furthermore, migration can represent an exceptionally stressful circumstance linked to various social disadvantages both in the country of origin and during resettlement, which can lead to the unraveling of a previously sufficiently compensated psychopathological picture [87]. Clarification of the role of migration in the development and detection of autism spectrum disorder can provide etiological insights and help reveal preventable and addressable health inequalities. Attention to the possibility of such health inequalities occurring is particularly important in Western countries and, even more so, in subjects at risk of misdiagnosis or late diagnosis, such as ASD females [88].

## 5. Limits

The present study should be considered in light of some limitations. First of all, the use of self-reported tools may lead to biases in which participants overestimate or underestimate their symptoms. Secondly, the therapies taken by the patients before and at the time of hospitalization may have influenced the overall clinical status at the time of the evaluation. Third, in daily clinical practice, it can be difficult to decide which adult patients may need and benefit from an ASD evaluation.

Considering these limitations and by virtue of the ease and speed of administration, it could be appropriate to provide such evaluation measures to those patients who report in their psychobiography elements attributable to autistic traits such as difficulties in relationships with peers, emotional dysregulation, or particularly intense interests. In light of recent scientific evidence, we also recommend providing this self-assessment tool to patients who have received multiple diagnoses of FEDs and BPD in comorbidity with other affective or anxiety disorders or with repeated histories of interpersonal trauma.

A prompt diagnosis can help women develop a positive sense of identity [55] and make supports like employment inclusion and disability benefits possible [89]. It can also counteract social criticism and offer a reason for actions, which lessens the sense of guilt, explains past experiences, and helps the person make sense of them [90]. The management of ASD typically entails techniques to lessen the influence of autistic qualities on day-to-day functioning, like offering the assistance and training required to function on one’s own [91]. This rehabilitation strategy seeks to improve the environment, for example, by making it more predictable and lowering the overload of sensory stimuli while also increasing physical comfort and lowering anxiety [92,93].

Behavioral and psychosocial therapy are two of the main strategies used for the care of autism [94,95]. In this context, it is noteworthy that repressing repetitive behaviors may have a negative effect, for they could be coping mechanisms for people with ASD [89]. As a result of the bullying and persecution that people on the autistic spectrum often experience, trauma-informed care is crucial [92]. Ultimately, although there is no medication that can treat autism per se, there are medications that can address its comorbidities, and because ASD seems to be a marker of unpredictable adverse responses to psychotropic drugs, medication introduction should be approached carefully and at moderate doses [93]. Additionally, because there is a chance of adverse effects unique to women, extra consideration should be given to gender when administering pharmaceutical treatment [96]. For example, aripiprazole, an atypical antipsychotic drug generally used to treat irritability in children with autism, is one of the few of its category that does not cause hyperprolactinemia, which can lead to gynecomastia and galactorrhea [97].

## 6. Conclusions

To this date, research about gender differences in the autism spectrum is still in its infancy, and currently, there exists no conclusive description of the female autism phenotype that could guide endeavors to mitigate the bias in ascertainment toward girls and women who have ASD [13].

Based on the examples of the two young girls that have been reported, we propose that the challenges associated with diagnosing ASD in females are a result of both the unique traits of the female autism phenotype and the characteristics of the diagnostic systems that are intended to recognize and support individuals with ASD. Specifically, we propose that girls who are particularly committed to and adept at masking and camouflaging are more likely to have an ASD that is not recognized and that girls’ propensity to experience internalizing problems but not externalizing ones is another risk factor for an ASD that is not recognized [55,88]. Under this approach, studies ought to concentrate on recognizing and fighting the diagnostic prejudice that prevents the recognition of female appearances of ASD.

In this framework, by delving deeper into the female autism phenotypes, the sex ratio in ASD prevalence may be reexamined. Additionally, a better understanding of several psychiatric conditions that are frequently diagnosed in females may be attained, and the issue of gender differences in psychiatry as a whole may be rethought in the context of a neurodevelopmental approach to psychopathology.

## Figures and Tables

**Table 1 brainsci-14-00037-t001:** Score and percentage score reported by X.Y. in the AdAS Questionnaire’s domains.

AdAS Spectrum Domains	Score	N° of Items	Percentage of Positive Items
Childhood/adolescence	14	21	66.7%
Verbal communications	9	18	50%
Nonverbal communications	18	28	64.3%
Empathy	6	12	50%
Inflexibility and adherence to routine	18	43	41.9%
Restricted interests and rumination	10	21	47.6%
Hyper–hypo reactivity to sensory input	2	17	11.8%
Total	77	160	48.1%

**Table 2 brainsci-14-00037-t002:** Score and percentage score reported by X.Y. in the CAT-Q Questionnaire’s areas.

CAT-Q Domains	Score	N° of Items
Compensation	23	9
Masking	43	8
Assimilation	44	8
Total	110	175

**Table 3 brainsci-14-00037-t003:** Score and percentage score reported by X.Y. in the EDI-2 Questionnaire’s domains.

EDI-2 Domains	Score	N° of Items
Drive towards thinness	13	7
Bulimia	4	7
Body dissatisfaction	11	9
Ineffectiveness	17	10
Perfectionism	8	6
Interpersonal distrust	18	7
Interoceptive Awareness	18	10
Maturity fears	11	8
Asceticism	9	8
Impulse Regulation	15	11
Social Insecurity	15	8
Total	139	273

**Table 4 brainsci-14-00037-t004:** Score and percentage score reported by W.Z. in the AdAS Questionnaire’s domains.

AdAS Spectrum Domains	Score	N° of Items	Percentage of Positive Items
Childhood/adolescence	11	21	52.4%
Verbal communications	8	18	44.4%
Nonverbal communications	18	28	62.3%
Empathy	3	12	25%
Inflexibility and adherence to routine	19	43	44.2%
Restricted interests and rumination	9	21	42.8%
Hyper-hypo reactivity to sensory input	5	17	29.4%
Total	73	160	45.6%

**Table 5 brainsci-14-00037-t005:** Score and percentage score reported by W.Z. in the TALS-SR Questionnaire’s domains.

TALS-SR Domains	Score	N° of Items	Percentage of Positive Items
Loss events	5	10	50%
Reactions to loss events	19	27	70.4%
Potentially traumatic events	9	20	45%
Reactions to loss or potentially traumatic events	4	18	22.2%
Re-experiencing	4	9	44.4%
Avoidance and numbing	6	11	54.5%
Maladaptive behaviors	7	8	87.5%
Arousal	1	5	20%
Personality traits and risk factors	5	7	71.4%
Total	60	116	51.7%

**Table 6 brainsci-14-00037-t006:** Score and percentage score reported by W.Z. in the CAT-Q Questionnaire’s areas.

CAT-Q Domains	Score	N° of Items
Compensation	16	9
Masking	33	8
Assimilation	33	8
Total	82	175

## Data Availability

Data are contained within the article.

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
