# Peer review of "May Female Autism Spectrum Be Masked by Eating Disorders, Borderline Personality Disorder, or Complex PTSD Symptoms? A Case Series"

_brainsci, 2023, doi:10.3390/brainsci14010037_

Round 1
Reviewer 1 Report
Comments and Suggestions for Authors
This is a very interesting paper discussing the need to further gender-specific research in the particular context of autism spectrum disorders. The authors have also presented clinical cases to illustrate this important topic. The paper is well written and of interest for the readers; however, several minor changes are needed to improve the paper.
ABSTRACT
I recommed to structure the abstract into introduction (currently done), aims, methods, results and conclusions. The methods could be the review on the topic and the case series. This should be clarified as a methods section.
INTRODUCTION
1- Some of the potential barriers to the clinical care of women suffering from autism spectrum disorders is the access to services and the lack of early detection programs. The vast majority of programs in mental health are focused on the care of first-episode of psychosis programs; however, several women with ASD are underdiagnosed. I recommend to expand these ideas in the introduction section.
METHODS
There is no methods section. I recommend to describe the selection of cases, and how these case are analysed. The introduction should be expanded according to the literature on the field of gender differences in ASD.
CASE PRESENTATIONS
The clinical relevance of psychobiography and early symptoms should be explained in the methods section. How this evaluation was those? The description of cases should be homogenized.
How migration is influencing the psychopathology and clinical course of women suffering from ASD?
Author Response
Reviewer 1:
This is a very interesting paper discussing the need to further gender-specific research in the particular context of autism spectrum disorders. The authors have also presented clinical cases to illustrate this important topic. The paper is well written and of interest for the readers; however, several minor changes are needed to improve the paper.
Response: we thank the reviewer for giving us the opportunity to improve our work.
ABSTRACT
I recommed to structure the abstract into introduction (currently done), aims, methods, results and conclusions. The methods could be the review on the topic and the case series. This should be clarified as a methods section.
Response: we thank the reviewer for the suggestion. We proceeded to modify the abstract and structure it as suggested. We also added the aims section and implemented the methods on as follows:
“Aims: we aimed to describe to describe a brief case in order to confirm the diagnostic difficulties that ASD female undergo during their clinical evaluation, and the possible alternative phenotype that they can manifest. Methods:We reported the cases of two young women on the autism spectrum that came to clinical attention only after the development of a severe symptomatology attributed to other mental disorders, overlooking the presence of underlying autism spectrum features and a brief résumé of the literature on the topic.”
INTRODUCTION
1- Some of the potential barriers to the clinical care of women suffering from autism spectrum disorders is the access to services and the lack of early detection programs. The vast majority of programs in mental health are focused on the care of first-episode of psychosis programs; however, several women with ASD are underdiagnosed. I recommend to expand these ideas in the introduction section.
Response: we thank the reviewer for the suggestion. We decided to better discuss the topic in the Introduction as follows:
“Ultimately, females are far more likely than males to have an undetected ASD, because their problems are sometimes mislabeled or overlooked completely [11]. As a result, many females who, upon a skilled assessment, would fully meet the diagnostic criteria for ASD, never receive a diagnosis or the possible assistance that accompanies it, and even when they are found they are diagnosed later than the male counterpart and receive less support in the meanwhile [49], indeed, the majority of programs in mental health are focused on the care of first-episode of psychosis which is a far more common male presentation. Moreover, teachers underreport autistic qualities in their female students, which in turn need more severe autistic symptoms and more cognitive and behavioral deficits than male students do in order to be recognized [25, 50, 51]. These prejudices are unavoidable, and they appear to have been particularly prevalent among those who are not experts in diagnosing neurodevelopmental disorders but who are nonetheless powerful gatekeepers to pertinent treatments for individuals with ASD [52, 53]. The lack of researched on this gender prejudice has finally been recognized as one of the main key problems in the autism community and has important implications for the health and well-being of girls and women on the spectrum [54, 55].”
METHODS
There is no methods section. I recommend to describe the selection of cases, and how these case are analysed. The introduction should be expanded according to the literature on the field of gender differences in ASD.
Response: we thank the reviewer for the suggestion. We added a Methods section describing how the cases were recruited and analyzed as follows:
“2. Methods:
The subjects were recruited from inpatients afferent at the Psychiatric Clinic of the University of Pisa and did not report any language or intellectual disabilities that made it difficult to complete the examinations. During the hospitalization they underwent daily observation and assessment from trained psychiatrist and were evaluated with the following self-report questionnaire:
2.1 Adult Autism Subthreshold Spectrum (AdAS Spectrum):
The AdAS Spectrum is a self-report questionnaire designed to assess the wide range of autism spectrum manifestations in adults who do not have language or intellectual disabilities. It consists of 160 dichotomous items organized into seven domains which explores Childhood and adolescence, Verbal communication, Nonverbal communication, Empathy, Inflexibility and Adherence to Routine, Restricted interests and rumination and Hyper- and Hyporeactivity to Sensory Input. The questionnaire showed great psychometric properties, excellent internal consistency for the total score (Kuder–Richardson's coefficient=.964), convergent validity with other dimensional measures of ASD and a diagnostic threshold score of 70 [56, 57].
2.2 Camouflaging Autistic Traits Questionnaire (CAT–Q):
The CAT-Q is a questionnaire developed by Hull et al. to assess the dimension of camouflaging behaviors, for which is also available an Italian version. Both versions showed great internal consistency with a Cronbach's alpha of 0.904 in the Italian version, and great test re-test reliability and convergent validity with other measures of ASD [58, 59]. The questionnaire consists of 25 items organized in a seven-point Likert scale and divided in three domains investigating Compensation, Masking and Assimilation behaviors
2.3 Eating Disorder Inventory 2 (EDI-2):
The EDI-2 is a self-report questionnaire that measures psychological traits and symptom clusters relevant to the assessment of individuals with eating disorders. It consists in 91 items rated on a 6-point scale from never to always organized in 11 domains such as Drive towards thinness, Bulimia, Body dissatisfaction, Ineffectiveness, Perfectionism, Interpersonal distrust, Interoceptive Awareness, Maturity fears, Asceticism, Impulse Regulation andSocial Insecurity. The EDI-2 is vastly used in FED researches and has been validated for both clinical and general populations [60].
2.4 The ORTO-15
The ORTO-15 is a self-report questionnaire used for the assessment of eating behaviors ascribable to Orthorexia Nervosa. It consists in 15 items rated on a 4-point Likert scale and recognized two validated threshold score of <35, which displayed greater specificity, and <40 which displayer a greater sensitivity [61, 62].
2.5 Trauma and Loss Spectrum – Self Report (TALS-SR):
The TALS-SR questionnaire investigates the lifetime experience of a range of traumatic events or losses as well as symptoms, behaviors and personological traits that might represent manifestations or risk factors for the development of a stress-related disorder. It consists in 116 dichotomous items organized in nine domains exploring Loss events, Rrief reactions, Potentially traumatic events, Reactions to losses or upsetting events, Re-experiencing, Avoidance and numbing, Maladaptive coping, Arousal and Personal characteristics/risk factors. The questionnaire showed overall great psychometric properties [63, 64].”
CASE PRESENTATIONS
The clinical relevance of psychobiography and early symptoms should be explained in the methods section. How this evaluation was those? The description of cases should be homogenized.
Response: we thank the reviewer for the suggestion. We proceeded to homogenize the description of the cases and added a brief paragraph in the Methods section explaining the clinical relevance of psychobiography and early symptoms as well as how their evaluation was carried as follows:
“Early symptoms that may have been present during childhood were assessed with the help of parents and through life-time self-report questionnaires. The evaluation of early symptoms and the provision of a comprehensive psychobiography appears particularly relevant in the evaluation of females with ASD. In fact, some typical characteristics of ASD such as deficits in verbal and non-verbal communication can subsequently be masked and therefore difficult to detect thanks to the acquisition of camouflaging strategies.”
How migration is influencing the psychopathology and clinical course of women suffering from ASD?
Response: we thank the reviewer for the question. We decided to address this timely topic in the discussion as follows:
“In this context, given the expanding incidence of ASD, the hunt for environmental risk factors is becoming more crucial. One such issue that has drawn attention from researchers is migration with low vitamin D levels and ethnicity proposed as potential explanatory processes [77, 78]. However, the quality, sample sizes, and definitions of autism used in prior research on the relationship between migration and the risk of ASD differ significantly and have shown contradictory results, such as higher, comparable and even lower risk of autism in migrant children [78 – 86]. Furthermore, migration can represent an exceptionally stressful circumstance, linked to various social disadvantages both in the country of origin and during resettlement, which can lead to the unraveling of a previously sufficiently compensated psychopathological picture [87]. Clarification of the role of migration in the development and detection of autism spectrum disorder can provide etiological insights and help reveal preventable and addressable health inequalities. Attention to the possibility of such health inequalities occurring is particularly important in Western countries and, even more so, in subjects at risk of misdiagnosis or late diagnosis such as ASD females [88].”
Reviewer 2 Report
Comments and Suggestions for Authors
This case series explores the conundrum of female autism spectrum disorders in relationship with eating disorders, borderline personality disorder, or complex PTSD. There are some methodological aspects that require supplementary discussions, and proofreading is needed. Please refer to the following observations:
Title
-“case series”
Abstract
-“female autistic phenotype”
Introduction
Line 34- what risks, specifically?
Line 37- “females with ASD”;
Line 38,46- the patient-first formulation is advisable, i.e., “females with ASD”;
Lines 38-39- please rephrase (“One proposed explanation…finds its explanation…”);
Line 94- all acronyms should be defined the first time they are used, in this case PTSD;
Line 95- “the complex PTSD”;
Line 99- a comma between “screened” and “leading” is needed;
Line 101- please define what “autism spectrum” means in the context of this manuscript. Is it represented by disorders of this spectrum (corresponding to DSM-5 criteria, for example), isolated clinical features (as defined in lines 67-70), or something else?
Case presentation
Lines 119-120- “food domain” or just “selective eating behaviors”
Line 129 – “symptoms” instead of “expressiveness” would sound more natural;
Line 138- did the Authors mean “free-floating anxiety”?
Line 143- what did the Authors mean by “which were at times criticized”? Did the patient have a variable insight into her thought disorders?
Including instruments for ASD in both of these patients’ evaluations is beneficial, of course, but it is unclear what their other diagnoses were, if any, at the time of examination in the Authors’ center. Maybe a structured interview, like MINI or SCID-5-CV, would have been helpful to exclude other potential comorbidities. Also, what happened to the BPD diagnosis previously established in another psychiatric center in both patients? Were there any confounding factors that could alter the scoring of the instruments used by the Authors? What about the impact of medications on the overall clinical status at the time of evaluation? The rationale for examining these two patients with instruments for ASD needs more elaboration. It is unclear when a clinician may recommend such testing and what practical recommendations the Authors may provide, based on their current research. These are limitations of the study and should be placed in the section dedicated to these aspects.
Line 177- what does “potus” mean in this context? If a substance use disorder was suspected, please specify so.
Please consider inserting a timeline for each patient, with pharmacological +/- psychotherapies used; doses and durations of each treatment should be mentioned, where available. Also, “gabapentinoids” is too vague, please specify the international non-proprietary name of each drug used, doses, and duration.
Line 196,205- “At the age of 20 (or 21)”;
Line 216- why are the patient’s initials changed here?
Discussions
Line 300- “and at now” is an improper formulation, please clarify;
What are the limitations and strengths of the current case series?
Any directions for further research based on the discoveries of this research?
Are there any therapeutic implications of this case series for clinicians?
Please remove lines 318-321, which look like remnants from the manuscript’s template.
Line 312- Are there no IRB approvals needed for such case reports by local legislation?
Comments on the Quality of English LanguageModerate English language editing is needed.
Author Response
Reviewer 2:
This case series explores the conundrum of female autism spectrum disorders in relationship with eating disorders, borderline personality disorder, or complex PTSD. There are some methodological aspects that require supplementary discussions, and proofreading is needed. Please refer to the following observations:
Response: we thank the reviewer for giving us the opportunity to improve our work.
Title
-“case series”
Response: we thank the reviewer for noticing the mistake, we proceeded to correct the text.
Abstract
-“female autistic phenotype”
Response: we thank the reviewer for noticing the mistake, we proceeded to correct the text.
Introduction
Line 34- what risks, specifically?
Response: we thank the reviewer for the question. We decided to delve deeper into the topic as follows:
“Individuals with ASD often face the risk of experiencing a series of behavioral, emotional, social and professional challenges such as maintaining romantic and friendship relationships, fitting into a work environment and adapting to the norms imposed by the sociocultural context in which they find themselves.”
Line 37- “females with ASD”;
Line 38,46- the patient-first formulation is advisable, i.e., “females with ASD”;
Response: we thank the reviewer for noticing the mistake, we proceeded to correct the text.
Lines 38-39- please rephrase (“One proposed explanation…finds its explanation…”);
Response: we thank the reviewer for the suggestion, we proceeded to rephrase the sentence as follows:
“One proposed explanation for the diagnostic bias against females with ASD finds its ground in…”
Line 94- all acronyms should be defined the first time they are used, in this case PTSD;
Response: we thank the reviewer for the suggestion, we proceeded to define all acronyms the first time they are used.
Line 95- “the complex PTSD”;
Line 99- a comma between “screened” and “leading” is needed;
Response: we thank the reviewer for noticing the mistake, we proceeded to correct the text.
Line 101- please define what “autism spectrum” means in the context of this manuscript. Is it represented by disorders of this spectrum (corresponding to DSM-5 criteria, for example), isolated clinical features (as defined in lines 67-70), or something else?
Response: we thank the reviewer for the question. We proceeded to better explain what we meant for “autistic spectrum” as follows:
“In this work, we reported the cases of two young women within the autistic spectrum, where by spectrum is meant the set of typical symptoms described by the DSM, atypical symptoms, clusters of subclinical symptoms, personality traits and related behavioral manifestations, […]”
Case presentation
Lines 119-120- “food domain” or just “selective eating behaviors”
Response: we thank the reviewer for noticing the mistake, we proceeded to correct the text.
Line 129 – “symptoms” instead of “expressiveness” would sound more natural;
Response: we thank the reviewer for noticing the mistake, we proceeded to correct the text.
Line 138- did the Authors mean “free-floating anxiety”?
Response: we thank the reviewer for noticing the mistake, we proceeded to correct the text.
Line 143- what did the Authors mean by “which were at times criticized”? Did the patient have a variable insight into her thought disorders?
Response: we thank the reviewer for the question, the phrase meant that the patient had a variable and discontinuous insight on her disorder and that at times, she recognized her beliefs as erroneous, unfounded and pathological. Upon you question we decided to better clarify in the text as follows:
“[…] the appearance ideas of reference and persecution and which ultimately evolved towards a delirium of substitution aimed at her parents and the resurgence of restrictive eating habits. Interestingly, during this time, the girl had a variable and discontinuous insight on her disorder and that at times, she recognized her beliefs as erroneous, unfounded and pathological.”
Including instruments for ASD in both of these patients’ evaluations is beneficial, of course, but it is unclear what their other diagnoses were, if any, at the time of examination in the Authors’ center. Maybe a structured interview, like MINI or SCID-5-CV, would have been helpful to exclude other potential comorbidities. Also, what happened to the BPD diagnosis previously established in another psychiatric center in both patients? Were there any confounding factors that could alter the scoring of the instruments used by the Authors? What about the impact of medications on the overall clinical status at the time of evaluation? The rationale for examining these two patients with instruments for ASD needs more elaboration. It is unclear when a clinician may recommend such testing and what practical recommendations the Authors may provide, based on their current research. These are limitations of the study and should be placed in the section dedicated to these aspects.
Response: we thank the reviewer for the questions. For both girls, the previous diagnosis of “Borderline Personality Disorder and Anorexia Nervosa” and “Borderline Personality Disorder and Substance abuse” were confirmed. Upon your suggestion, we decided to add a Limits section as follows:
“The present study should be considered in light of some limitations. First of all, the use of self-reported tools may lead to biases in which participants overestimate or underestimate their symptoms. Secondly, the therapies taken by the patients before and at the time of hospitalization may have influenced the overall clinical status at the time of the evaluation. Third, in daily clinical practice it can be difficult to decide which adult patients may need and benefit from an ASD evaluation. Considering these limitations and by virtue of the ease and speed of administration, it could be appropriate to provide such evaluation measures to those patients who report in their psychobiography elements attributable to autistic traits such as difficulties in relationships with peers, emotional dysregulation or particularly intense interests. In light of recent scientific evidence, we also recommend providing this self-assessment tool to patients who have received multiple diagnosed of FEDs and BPD in comorbidity with other affective or anxiety disorders or with repeated histories of interpersonal trauma.”
Line 177- what does “potus” mean in this context? If a substance use disorder was suspected, please specify so.
Response: we thank the reviewer for the question, we decided to change “potus” with “alcohol use disorder”.
Please consider inserting a timeline for each patient, with pharmacological +/- psychotherapies used; doses and durations of each treatment should be mentioned, where available. Also, “gabapentinoids” is too vague, please specify the international non-proprietary name of each drug used, doses, and duration.
Response: we thank the reviewer for the suggestion. We proceeded to change “gabapentinoids” whit “Pregabalin” and to include doses and duration of the therapy taken, where available.
Line 196,205- “At the age of 20 (or 21)”;
Response: we thank the reviewer for noticing the mistake, we proceeded to correct the text.
Line 216- why are the patient’s initials changed here?
Response: we thank the reviewer for noticing the mistake, we proceeded to correct the text.
Discussions
Line 300- “and at now” is an improper formulation, please clarify;
Response: we thank the reviewer for the suggestion, we proceeded to change “at now” with “currently”
What are the limitations and strengths of the current case series?
Response: we thank the reviewer for the questions, we proceeded to add a section assessing the topic as follows:
“The present study should be considered in light of some limitations. First of all, the use of self-reported tools may lead to biases in which participants overestimate or underestimate their symptoms. Secondly, the therapies taken by the patients before and at the time of hospitalization may have influenced the overall clinical status at the time of the evaluation. Third, in daily clinical practice it can be difficult to decide which adult patients may need and benefit from an ASD evaluation. Considering these limitations and by virtue of the ease and speed of administration, it could be appropriate to provide such evaluation measures to those patients who report in their psychobiography elements attributable to autistic traits such as difficulties in relationships with peers, emotional dysregulation or particularly intense interests. In light of recent scientific evidence, we also recommend providing this self-assessment tool to patients who have received multiple diagnosed of FEDs and BPD in comorbidity with other affective or anxiety disorders or with repeated histories of interpersonal trauma.”
Any directions for further research based on the discoveries of this research?
Response: we thank the reviewer for the question. We decided to address the topic in the Conclusion section as follows:
“Based on the examples of the two young girls that have been reported, we propose that the challenges associated with diagnosing ASD in females are a result of both the unique traits of the female autism phenotype and the characteristics of the diagnostic systems that are intended to recognize and support individuals with ASD. Specifically, we propose that girls who are particularly committed to and adept at masking and camouflaging are more likely to have an ASD that is not recognized, and that girls' propensity to experience internalizing problems but not externalizing ones is another risk factor for an ASD that is not recognized [52, 55]. Under this approach, studies ought to concentrate on recognizing and fighting the diagnostic prejudice that prevents the recognition of female appearances of ASD.”
Are there any therapeutic implications of this case series for clinicians?
Response: we thank the reviewer for the question. We decided to add a paragraph addressing the therapeutic implications in the Discussion section as follows:
“A prompt diagnosis can help women develop a positive sense of identity [55] and make supports like employment inclusion and disability benefits possible [89]. It can also counteract social criticism and offer a reason for actions, which lessens the sense of guilt as well as explain past experiences and help the person make sense of them [90]. The management of ASD typically entails techniques to lessen the influence of the autistic qualities on day-to-day functioning, like offering the assistance and training required to function on one's own [91]. This rehabilitation strategy seeks to improve the environment, for example by making it more predictable and lowering the overload of sensory stimuli, while also increasing physical comfort and lowering anxiety [92, 93].
Behavioral and psychosocial therapy are two of the main strategies used for the care of autism [94, 95]. In this context, it is noteworthy that repressing repetitive behaviors may have a negative effects for they could be coping mechanisms for people with ASD [89]. As a result of the bullying and persecution that people on the autistic spectrum often experience, trauma-informed care is crucial [A40]. Ultimately, although there is no medication that can treat autism per se, there are medications that can address its co-morbidities, and because ASD seems to be a marker of unpredictable adverse responses to psychotropic drugs, medication introduction should be done carefully and at moderate doses [93]. Additionally, because there is a chance of adverse effects unique to women, extra consideration should be given to gender when administering pharmaceutical treatment [96]. For example, aripiprazole, an atypical antipsychotic drugs generally used to treat irritability in children with autism, is one of the few of its category that does not cause hyperprolactinemia, which can lead to gynecomastia and galactorrhea [97].”
Please remove lines 318-321, which look like remnants from the manuscript’s template.
Response: we thank the reviewer for the suggestion. Lines 318-321 represent the Authors contributions section requested from the journal. We proceeded to remove line 331-332 which were remnants from the manuscript’s template.
Line 312- Are there no IRB approvals needed for such case reports by local legislation?
Response: we thank the reviewer for the question. IRB approval is not needed by our local legislation for the description of clinical cases. However, it is requested an informed consent form signed by both subjects which was provided to the editor upon the submission of the manuscript.
Reviewer 3 Report
Comments and Suggestions for Authors
The manuscript makes a valuable contribution to the understanding of misdiagnosis in females with ASD, shedding light on the role of camouflaging behaviors and the impact of gender-specific phenotypes. With some additional refinements, particularly in the areas of diagnostic criteria analysis, case study integration, and discussion on intervention strategies, the manuscript has the potential to be a significant and insightful piece in the field of autism research.
The comments are as follows:
1. The cases of X.Y. and W.Z. are mentioned, but the manuscript could benefit from a more detailed exploration of their individual experiences. Providing specific examples of camouflaging behaviors observed in these cases would not only illustrate the points made but also make the content more relatable to readers. A deeper dive into the nuances of each case would strengthen the manuscript's overall impact.
2. While the manuscript effectively highlights the potential negative outcomes of misdiagnosis, there is a notable absence of discussion on intervention strategies. Including a brief discussion on targeted interventions tailored to address the unique challenges faced by females with ASD would provide a more balanced perspective. This addition would enhance the manuscript's practical relevance for clinicians and researchers.
Author Response
Reviewer 3:
The manuscript makes a valuable contribution to the understanding of misdiagnosis in females with ASD, shedding light on the role of camouflaging behaviors and the impact of gender-specific phenotypes. With some additional refinements, particularly in the areas of diagnostic criteria analysis, case study integration, and discussion on intervention strategies, the manuscript has the potential to be a significant and insightful piece in the field of autism research.
Response: we thank the reviewer for giving us the opportunity to improve our work.
The comments are as follows:
- The cases of X.Y. and W.Z. are mentioned, but the manuscript could benefit from a more detailed exploration of their individual experiences. Providing specific examples of camouflaging behaviors observed in these cases would not only illustrate the points made but also make the content more relatable to readers. A deeper dive into the nuances of each case would strengthen the manuscript's overall impact.
Response: we thank the reviewer for the suggestion. We decided to better describe the camouflaging behaviors adopted by the patients in the text, including extensive examples as follows:
X.Y. : “During the clinical interview, the camouflaging strategies implemented by the patient were explored in depth. In particular, she reported to have a series of clothes and accessories specifically designed to appear appropriate to the situation in which they would be worn, for example clothes for the library, for school or for the bar. She also reported that she remembered, through the questionnaire, how during childhood she had committed herself to studying facial expressions and non-verbal communicative movements, trying to copy those seen by children at school in various situations. Finally reports making a conscious effort during conversations to maintain a balance between talking and listening, and to fill the gaps with verbal encouragers.”
W.Z. “During the clinical interview, the camouflaging strategies implemented by the patient were explored in depth. In particular, she reported to actively try to reduce her fidgeting and stimming movements and making an effort to intentionally maintain eye contact or the appearance of eye contact, for example by looking at the interlocutor's forehead. She also reports that she is used to smiling when someone is talking to her because she has learned over time that it makes her appear less rude and more approachable.”
- While the manuscript effectively highlights the potential negative outcomes of misdiagnosis, there is a notable absence of discussion on intervention strategies. Including a brief discussion on targeted interventions tailored to address the unique challenges faced by females with ASD would provide a more balanced perspective. This addition would enhance the manuscript's practical relevance for clinicians and researchers.
Response: we thank the reviewer for the question. We decided to add a paragraph addressing the therapeutic implications in the Discussion section as follows:
“A prompt diagnosis can help women develop a positive sense of identity [55] and make supports like employment inclusion and disability benefits possible [89]. It can also counteract social criticism and offer a reason for actions, which lessens the sense of guilt as well as explain past experiences and help the person make sense of them [90]. The management of ASD typically entails techniques to lessen the influence of the autistic qualities on day-to-day functioning, like offering the assistance and training required to function on one's own [91]. This rehabilitation strategy seeks to improve the environment, for example by making it more predictable and lowering the overload of sensory stimuli, while also increasing physical comfort and lowering anxiety [92, 93].
Behavioral and psychosocial therapy are two of the main strategies used for the care of autism [94, 95]. In this context, it is noteworthy that repressing repetitive behaviors may have a negative effect for they could be coping mechanisms for people with ASD [89]. As a result of the bullying and persecution that people on the autistic spectrum often experience, trauma-informed care is crucial [92]. Ultimately, although there is no medication that can treat autism per se, there are medications that can address its co-morbidities, and because ASD seems to be a marker of unpredictable adverse responses to psychotropic drugs, medication introduction should be done carefully and at moderate doses [93]. Additionally, because there is a chance of adverse effects unique to women, extra consideration should be given to gender when administering pharmaceutical treatment [96]. For example, aripiprazole, an atypical antipsychotic drug generally used to treat irritability in children with autism, is one of the few of its category that does not cause hyperprolactinemia, which can lead to gynecomastia and galactorrhea [97].”
Reviewer 4 Report
Comments and Suggestions for Authors
This manuscript summarized the results of studies on subthreshold autism spectrum features in females and described two highly-suspected ASD cases with the symptoms of borderline personality traits, eating disorder, and substance use. Given that gender-specific clinical manifestations of autism spectrum disorder warrant further investigation, this case report added knowledge to the field and has the clinical contribution.
I would like to suggest the authors revised their manuscript in some ways. First, the lengths of the paragraphs in the current manuscript are too long to be followed. For example, the first paragraph of Introduction section has 51 lines (line 29-79). The authors may consider split it into three paragraphs based on the themes. The lengths of descriptions of the two cases were also very long. It hinders the readers from catching the important meanings of the descriptions.
Second, several sentences warrant revisions. For example, line 71-73, 182-186.
Third, English writing warrants revisions. For example, the word “indeed” appeared in the manuscript for several times.
Fourth, typos warrant being detected and corrected thorough the manuscript.
Comments on the Quality of English LanguageEnglish writing warrants revisions.
Author Response
Reviewer 4:
This manuscript summarized the results of studies on subthreshold autism spectrum features in females and described two highly-suspected ASD cases with the symptoms of borderline personality traits, eating disorder, and substance use. Given that gender-specific clinical manifestations of autism spectrum disorder warrant further investigation, this case report added knowledge to the field and has the clinical contribution.
Response: we thank the reviewer for giving us the opportunity to improve our work.
I would like to suggest the authors revised their manuscript in some ways. First, the lengths of the paragraphs in the current manuscript are too long to be followed. For example, the first paragraph of Introduction section has 51 lines (line 29-79). The authors may consider split it into three paragraphs based on the themes. The lengths of descriptions of the two cases were also very long. It hinders the readers from catching the important meanings of the descriptions.
Response: we thank the reviewer for the suggestion. We proceeded to divide the text and the description of the cases in smaller paragraphs based on themes as suggested.
Second, several sentences warrant revisions. For example, line 71-73, 182-186.
Response: we thank the reviewer for the suggestion. We proceed to deeply revise the text.
Third, English writing warrants revisions. For example, the word “indeed” appeared in the manuscript for several times.
Response: we thank the reviewer for the suggestion, we have subjected the text to an English language revision.
Fourth, typos warrant being detected and corrected thorough the manuscript.
Response: we thank the reviewer for the suggestion, we have subjected the text to an extensive revision.
Reviewer 5 Report
Comments and Suggestions for Authors
The manuscript entitled “May female autism spectrum be masked by eating disorders, borderline personality disorder or complex PTSD symptoms? A case serie” reports two clinical cases of mental disorders with diagnosed eating problems, personality and affective disorders. The authors suggested that these girls could be misdiagnosed due to the presence of comorbid autistic traits evaluated with AdAS and CAT-Q questionnaires. Therefore, the observations and conclusions made by the authors represent a primary step for future reconsideration of autism diagnosis in females based on clinical symptoms of “female autism phenotype”. The manuscripr is a well-written and well-structured article, which emphasizes the necessity of a deeper insight into the examination of comorbidity of mental disorders and symptoms in females.
The use of AdAS and CAT-Q questionnaires for the evaluation of autism subthreshold disorder has been in detailed explained by the authors. I have only one minor suggestion to the authors to add a value of internal consistency and validity (i.e. Cronbach’s alpha) for the used questionnaires (CAT-Q, AdAS), which was previously reported for a relevant cohort (for example, in Introduction).
Finally, there are some minor mistakes in spelling, the authors have to check a text for them (i.e. Line 129 - check for the presence of “he” instead of “she”).
Comments on the Quality of English Language
Minor English editing has to be performed; however, English is fine in general.
Author Response
Reviewer 5:
The manuscript entitled “May female autism spectrum be masked by eating disorders, borderline personality disorder or complex PTSD symptoms? A case serie” reports two clinical cases of mental disorders with diagnosed eating problems, personality and affective disorders. The authors suggested that these girls could be misdiagnosed due to the presence of comorbid autistic traits evaluated with AdAS and CAT-Q questionnaires. Therefore, the observations and conclusions made by the authors represent a primary step for future reconsideration of autism diagnosis in females based on clinical symptoms of “female autism phenotype”. The manuscripr is a well-written and well-structured article, which emphasizes the necessity of a deeper insight into the examination of comorbidity of mental disorders and symptoms in females.
Response: we thank the reviewer for giving us the opportunity to improve our work.
The use of AdAS and CAT-Q questionnaires for the evaluation of autism subthreshold disorder has been in detailed explained by the authors. I have only one minor suggestion to the authors to add a value of internal consistency and validity (i.e. Cronbach’s alpha) for the used questionnaires (CAT-Q, AdAS), which was previously reported for a relevant cohort (for example, in Introduction).
Response: we thank the reviewer for the comment. Upon the suggestion we decided to add a brief description of the questionnaires used as well as their value of internal consistency and validity as follows:
“2.1 Adult Autism Subthreshold Spectrum (AdAS Spectrum):
The AdAS Spectrum is a self-report questionnaire designed to assess the wide range of autism spectrum manifestations in adults who do not have language or intellectual disabilities. It consists of 160 dichotomous items organized into seven domains which explores Childhood and adolescence, Verbal communication, Nonverbal communication, Empathy, Inflexibility and Adherence to Routine, Restricted interests and rumination and Hyper- and Hyporeactivity to Sensory Input. The questionnaire showed great psychometric properties,cexcellent internal consistency for the total score (Kuder–Richardson's coefficient=.964), convergent validity with other dimensional measures of ASD and a diagnostic threshold score of 70 [56, 57].
2.2 Camouflaging Autistic Traits Questionnaire (CAT–Q):
The CAT-Q is a questionnaire developed by Hull et al. to assess the dimension of camouflaging behaviors, for which is also available an Italian version. Both versions showed great internal consistency with a Cronbach's alpha of 0.904 in the Italian version, and, test re-test reliability and convergent validity with other measures of ASD [58, 69]. The questionnaire consists of 25 items organized in a seven-point Likert scale and divided in three domains investigating Compensation, Masking and Assimilation behaviors.”
Finally, there are some minor mistakes in spelling, the authors have to check a text for them (i.e. Line 129 - check for the presence of “he” instead of “she”).
Response: we thank the reviewer for the suggestion, we have subjected the text to an extensive revision.
Round 2
Reviewer 2 Report
Comments and Suggestions for Authors
Thank you for considering my suggestions presented in the previous review round. However, some additional observations need to be addressed; please see below:
Line 18- Please start the sentence with a capital letter; „to describe” is mentioned twice in this sentence;
Line 109- „later than their male counterpart”;
Line 112- what does „autistic qualities” mean in this context? Did the Authors mean „autistic features” or „ autistic manifestations”?
Line 117- „lack of research”;
Line 122- „DSM-5”
Lines 166-170- it is unclear why an instrument for scoring orthorexia nervosa was chosen for these patients. All the variables monitored should be integrated theoretically, and the objectives of the paper need to be operationalized so that all the variables and associated instruments have a clear connection to the rationale of the paper.
Line 221- „that at times” is incorrect grammar; please revise;
Line 244- it is not clear why ORTO-15 was administered in this patient because the current mental status at the hospital admission and the patient’s personal history, as reported by the Authors, do not suggest the existence of orthorexia nervosa features; however, psychotic symptoms were present, and a scale for the evaluation of these manifestations was not recommended; please clarify for the readers the choice of the instruments used;
Line 413- „more crucial” is redundant, please rephrase.
Comments on the Quality of English LanguageMinor editing of English language is needed
Author Response
Reviewer 2:
Thank you for considering my suggestions presented in the previous review round. However, some additional observations need to be addressed; please see below:
Response: we thank the reviewer for giving us the opportunity to improve our work.
Line 18- Please start the sentence with a capital letter; „to describe” is mentioned twice in this sentence;
Response: we thank the reviewer for noticing the mistake, we proceeded to correct the text.
Line 109- „later than their male counterpart”;
Response: we thank the reviewer for noticing the mistake, we proceeded to correct the text.
Line 112- what does „autistic qualities” mean in this context? Did the Authors mean „autistic features” or „ autistic manifestations”?
Response: we thank the reviewer for the suggestion. We proceeded to change the text in “ autistic features”.
Line 117- „lack of research”;
Response: we thank the reviewer for noticing the mistake, we proceeded to correct the text.
Line 122- „DSM-5”
Response: we thank the reviewer for noticing the mistake, we proceeded to correct the text.
Lines 166-170- it is unclear why an instrument for scoring orthorexia nervosa was chosen for these patients. All the variables monitored should be integrated theoretically, and the objectives of the paper need to be operationalized so that all the variables and associated instruments have a clear connection to the rationale of the paper.
Response: we thank the reviewer for the comment. We decided to add a brief paragraph explaining our reason behind the inclusion of an instrument for the assessment of orthorexia nervosa as follows:
“The choice to include the ORRTO-15 in the assessment of the first patient was guided by the increasing body of researches that are confirming important symptomatologic overlap between ON and AN. Specifically, ON similarities to AN are sparking discussion on whether the first is a distinct disorder or a subtype of AN that parallels the growth of the modern ideal of healthy eating, which is progressively displacing the ideal of thinness that was prevalent in the 1980s and 1990s. In addition to the possibility for dramatic weight loss, ON and AN also share high levels of anxiety, a strong drive to exercise control, and a tendency toward perfectionism.”
Line 221- „that at times” is incorrect grammar; please revise;
Response: we thank the reviewer for the suggestion. We proceeded to change the text in “sometimes”.
Line 244- it is not clear why ORTO-15 was administered in this patient because the current mental status at the hospital admission and the patient’s personal history, as reported by the Authors, do not suggest the existence of orthorexia nervosa features; however, psychotic symptoms were present, and a scale for the evaluation of these manifestations was not recommended; please clarify for the readers the choice of the instruments used;
Response: we thank the reviewer for the comment. We noticed that a part X.Y. clinical assessmt that led us to use the ORTO-15 was missing. We added it in the text as follows:
“During hospitalization, the patient continued to show a hyporexic eating pattern which was associated with concerns about the number of calories consumed but also about the quality of the food served, the use of dyes, preservatives and palm oils, avoiding red meats and sausages and preferring vegetables deemed "anti-inflammatory and anticar-cinogenic".”
Line 413- „more crucial” is redundant, please rephrase.
Response: we thank the reviewer for the suggestion. We proceeded to change the text in “increasingly relevant”.